# TOAST: TOPOLOGICAL ALGORITHM FOR SINGULARITY TRACKING

## ABSTRACT

The manifold hypothesis, which assumes that data lie on or close to an unknown manifold of low intrinsic dimensionality, is a staple of modern machine learning research. However, recent work has shown that real-world data exhibit distinct non-manifold structures, which result in singularities that can lead to erroneous conclusions about the data. Detecting such singularities is therefore crucial as a precursor to interpolation and inference tasks. We address detecting singularities by developing (i) *persistent local homology*, a new topology-driven framework for quantifying the intrinsic dimension of a data set locally, and (ii) *Euclidicity*, a topology-based multi-scale measure for assessing the 'manifoldness' of individual points. We show that our approach can reliably identify singularities of complex spaces, while also capturing singular structures in real-world data sets.

## 1 INTRODUCTION

The ever-increasing amount and complexity of real-world data necessitate the development of new methods to extract less complex—but still *meaningful*—representations of the underlying data. One approach to this problem is via dimensionality reduction techniques, where the data is assumed to be of strictly lower dimension than its number of features. Traditional algorithms in this field such as PCA are restricted to linear descriptions of data, and are therefore of limited use for complex, non-linear data sets that often appear in practice. By contrast, non-linear dimensionality reduction algorithms, such as UMAP (McInnes et al., 2018), $t$-SNE (van der Maaten & Hinton, 2008), or autoencoders (Kingma & Welling, 2019) share one common assumption: the underlying data is supposed to be close to a manifold with small intrinsic dimension, i.e. while the input data may have a large ambient dimension $N$, there is a $n$-dimensional manifold with $n \ll N$ that best describes the data. For some data sets, this *manifold hypothesis* is appropriate: certain natural images are known to be well-described by a manifold, for instance (Carlsson, 2009), enabling the use of specialised autoencoders for visualisation (Moor et al., 2020). However, recent research shows evidence that the manifold hypothesis does not necessarily hold for complex data sets (Brown et al., 2022), and that manifold learning techniques tend to fail for non-manifold data (Rieck & Leitte, 2015; Scoccola & Perea, 2022). These failures are typically the result of *singularities*, i.e. regions of a space that violate the properties of a manifold. For example, the 'pinched torus,' an object obtained by compressing a neighbourhood of a random point in a torus to a single point, fails to satisfy the manifold hypothesis at the 'pinch point:' this point, unlike all other points of the 'pinched torus,' does *not* have a neighbourhood homeomorphic to $\mathbb{R}^2$ (see Fig. 1 for an illustration).

Since singularities—unlike outliers that arise from incorrect labels, for example—may carry relevant information (Jakubowski et al., 2020), we address the shortcomings of existing dimensionality reduction methods by assuming an agnostic view on any given data set. Instead of trying to prescribe the rigid requirements of a manifold, we consider intrinsic dimensionality to be a fundamentally *local phenomenon*: we permit dimensionality to vary across points in the data set, and, more importantly, across the *scale* of locality to be considered. The only assumption we make is that the data is of significantly lower dimension than the dimension of the ambient space. This perspective enables us to assess the deviation of individual points from idealised non-singular spaces, resulting in a measure of the *Euclidicity* of a point. Our method is based on a local version of topological data analysis (TDA), a method from computational topology that is capable of quantifying the shape of a data set on multiple scales (Edelsbrunner & Harer, 2010).

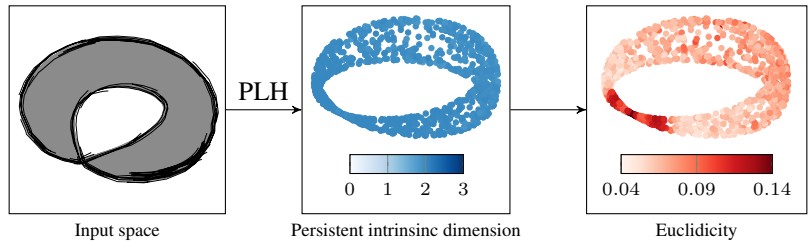

Figure 1: Overview of our method. Using *persistent local homology* (PLH), we derive a *persistent intrinsic dimension* and, subsequently, a *Euclidicity* score that measures the deviation from a space to a Euclidean model space. Here, *Euclidicity* highlights the singularity at the 'pinch point.' Please refer to Section 4 for more details.

**Our contributions.** We present a *universal framework for detecting singular regions in data.* This framework is agnostic with respect to geometric or stochastic properties of the underlying data and only requires a notion of intrinsic dimension of neighbourhoods. Our approach is based on a novel formulation of persistent local homology (PLH), a multi-parameter tool that detects the shape of local neighbourhoods of a given point in the data set, making use of multiple scales of locality. We employ PLH in two different capacities: (i) We use PLH to estimate the intrinsic dimension of a point locally. This enables us to assess how complex a given data set is, both in terms of the magnitude of the intrinsic dimension and in terms of the variance of its intrinsic dimension across individual points. (ii) Given the intrinsic dimension of the neighbourhood of a point, we use PLH to measure *Euclidicity*, a novel quantity that we define to measure the deviation of a point from being Euclidean. We also provide theoretical guarantees on the approximation quality for certain classes of spaces and show the utility of our proposed method experimentally on several data sets.

## 2 BACKGROUND: PERSISTENT HOMOLOGY AND STRATIFIED SPACES

We first provide an overview of persistent homology and stratified spaces, as well as their relation to *local homology*. The former concept constitutes a generic framework for assessing complex data at multiple scales by measuring its topological characteristics such as 'holes' and 'voids,' while the latter will subsequently serve as a general setting to describe singularities, in which our framework admits advantageous properties.

**Persistent homology.** Persistent homology is a method for computing topological features at different scales, capturing an intrinsic notion of relevance in terms of spatial scale parameters. Given a finite metric space $(\mathbb{X}, \mathrm{d})$, the *Vietoris–Rips complex* at step $t$ is defined as the abstract simplicial complex $\mathcal{V}(\mathbb{X}, t)$, in which an abstract $k$-simplex $(x_0, \ldots, x_k)$ of points in $\mathbb{X}$ is spanned if and only if $\mathrm{d}(x_i, x_j) \leq t$ for all $0 \leq i \leq j \leq k$.[1] For $t_1 \leq t_2$, the inclusions $\mathcal{V}(\mathbb{X}, t_1) \hookrightarrow \mathcal{V}(\mathbb{X}, t_2)$ yield a filtration, i.e. a sequence of nested simplicial complexes, which we denote by $\mathcal{V}(\mathbb{X}, \bullet)$. Applying the $i$th homology functor to this collection of spaces and inclusions between them induces maps on the homology level $f_i^{t_1, t_2} \colon \mathrm{H}_i(\mathcal{V}(\mathbb{X}, t_1)) \to \mathrm{H}_i(\mathcal{V}(\mathbb{X}, t_2))$ for any $t_1 \leq t_2$. The $i$th *persistent homology (PH)* of $\mathbb{X}$ with respect to the Vietoris-Rips construction is defined to be the collection of all these $i$th homology groups, together with the respective induced maps between them, and denoted by $\mathrm{PH}_i(\mathbb{X}; \mathcal{V})$. PH can therefore be viewed as a tool that keeps track of topological features such as holes and voids on multiple scales. For a more comprehensive introduction to PH in the context of machine learning, see Hensel et al. (2021). The so-called 'creation' and 'destruction' times of these features are summarised in a *persistence diagram* $\mathcal{D} \subset \mathbb{R} \times \mathbb{R} \cup \{\infty\}$, where any point $(b, d) \in \mathcal{D}$ corresponds to a homology class that arises at filtration step $b$, and lasts until filtration step $d$. The difference $|d - b|$ is referred to as the lifetime or eponymous *persistence* of this homology class. There are several distance measures for comparing persistence diagrams, one of them being the bottleneck distance $\mathrm{d}_{\mathrm{B}}$. For two persistence diagrams $\mathcal{D}, \mathcal{D}'$, it is defined as $\mathrm{d}_{\mathrm{B}}(\mathcal{D}, \mathcal{D}') := \inf_\gamma \sup_{x \in \mathcal{D}} \|x - \gamma(x)\|_\infty$, where $\gamma$ ranges over all bijections between $\mathcal{D}$ and $\mathcal{D}'$.

---

[1] For readers familiar with persistent homology, we depart from the usual convention of using $\epsilon$ as the threshold parameter since we will require it to denote the scale of our persistent local homology calculations.

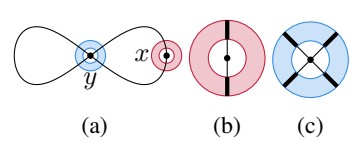

Figure 2: (a): Non-manifold space. (b): Annulus around a regular point $x$. (c): Annulus around a singular point. The neighbourhood around $y$ is different from all others.

**Stratified spaces.** Manifolds are widely studied and particularly well-behaved topological spaces: they locally resemble Euclidean space near any point. However, spaces that arise naturally often violate this local homogeneity condition, for example due to the occurrence of singularities (see Fig. 2 for an example), or since the space is of mixed dimensions. *Stratified spaces* generalise the concept of a manifold such that singular spaces are also addressed. Large classes of singular spaces can be formulated as stratified spaces, including (i) complex algebraic varieties, (ii) spaces that are disjoint unions of a finite number of manifolds of arbitrary dimensions, and (iii) spaces that admit isolated singularities. Being thus intrinsically capable of describing a wider class of spaces, we argue that stratified spaces are the right tool to analyse real-world data. Subsequently, we define stratified spaces in the setting of simplicial complexes. A stratified simplicial complex[2] of dimension $0$ is a finite set of points with the discrete topology. A stratified simplicial complex of dimension $n$ is an $n$-dimensional simplicial complex $X$, together with a filtration of closed subcomplexes $X = X_n \supset X_{n-1} \supset X_{n-2} \supset \cdots \supset X_{-1} = \emptyset$ such that $X_i \setminus X_{i-1}$ is an $i$-dimensional manifold for all $i$, and such that every point $x \in X$ possesses a distinguished local neighbourhood $U \cong \mathbb{R}^k \times c^\circ L$ in $X$, where $L$ is a compact stratified simplicial complex of dimension $n - k - 1$ and $c^\circ$ refers to the open cone construction (see Appendix A.1). If $X$ is a manifold, then independently of the point under consideration, $L$ is given by a sphere since for a manifold, *any* point admits a local neighbourhood that is homeomorphic to $\mathbb{R}^n$. This observation will serve as the primary motivation for our *Euclidicity* measure in Section 4.2.

**Local homology.** Local homology serves as a tool to quantify topological properties of infinitesimal small neighbourhoods of a fixed point. For a topological space $X$ and $x \in X$, its $i$th local homology group is defined as $\mathrm{H}_i(X, X \setminus x) := \varinjlim_U \mathrm{H}_i(X, X \setminus U)$, where the direct system is given by the induced maps on homology that arise via the inclusion of (small) neighbourhoods of $x$.[3] When $X$ is a simplicial complex, we may view $x$ as a vertex in $X$, using subdivision if necessary. Its *star* $\mathrm{St}(x)$ is defined to be the union of simplices in $X$ that have $x$ as a face, whereas its *link* $\mathrm{Lk}(x)$ consists of all simplices in $\mathrm{St}(x)$ that do not have $x$ as a face. Using excision and the long exact homology sequence (see Appendix A.3), we have

$$\mathrm{H}_i(X, X \setminus x) \cong \tilde{\mathrm{H}}_{i-1}(\mathrm{Lk}(x)). \tag{1}$$

The **key takeaway** here is that the homology of $\mathrm{Lk}(x)$ will usually differ from the homology of a sphere, once $\mathrm{Lk}(x)$ is not homotopy-equivalent to a sphere. For example, when $x$ is an isolated singularity in a stratified simplicial complex $X$ of dimension $n$, then its distinguished neighbourhood is given by $U \cong c^\circ L$. Thus, $\mathrm{Lk}(x) = L$ and $\mathrm{H}_i(X, X \setminus x) = \tilde{\mathrm{H}}_{i-1}(L)$ by Eq. (1), which is usually different from $\tilde{\mathrm{H}}_{i-1}(S^{n-1})$, when $x$ does not admit a Euclidean neighbourhood. This observation motivates and justifies using local homology for detecting non-Euclidean neighbourhoods.

## 3 RELATED WORK

Methods from topological data analysis have recently attracted much attention in machine learning, particularly due to persistent homology, which captures global topological properties of the underlying data set on different scales. We give a brief overview of existing methods in the emerging field of topology-driven singularity detection, outlining the differences to our approach below. Several works already assume a local perspective on homology to detect information about the intrinsic dimensionality of the data or the presence of certain singularities. Rieck et al. (2020) define persistent intersection homology via known stratifications, whereas Fasy & Wang (2016) and Bendich (2008), for instance, both present persistent versions of local homology. By contrast, Stolz et al. (2020) follow a different approach, where local homology is approximated as the absolute homology of

---

[2]Here, we actually mean the *geometric realisation* of the corresponding simplicial complex; by abuse of notation we may denote both objects by the term 'simplicial complex.'

[3]Heuristically, a local homology class can be thought of as a homology class of an infinitesimal small punctured neighbourhood of a point.

128 a small annulus of a given neighbourhood, resulting in an algorithm for geometric anomaly detec-
129 tion (which requires knowing the intrinsic dimension of the data set). Bendich et al. (2007) employ
130 persistence vineyards, i.e. continuous families of persistence diagrams, to assess the local homology
131 of a point in a stratified space, whereas Dey et al. (2014) use local homology to estimate the (global)
132 intrinsic dimension of hidden, possibly noisy manifolds. While manifold learning is concerned with
133 the development of algorithms that extract geometric information under the assumption that the
134 given data lie on a manifold, Brown et al. (2022) recently introduced the idea to assume data spaces
135 to consist of a *union of manifolds*. Intrinsic dimension is thus allowed to vary across connected
136 components of the data space, but singularities are excluded under this assumption, whereas our
137 framework detects the correct intrinsic dimension for large classes of singular spaces. Birdal et al.
138 (2021) define a global persistent homology dimension for describing neural networks; our persistent
139 intrinsic dimension, by contrast, is *local* and may thus change across different points in the data set.

140 **Key differences to existing approaches.** Our approach crucially differs from existing approaches
141 in essential components. In comparison to all aforementioned contributions, we capture additional
142 local geometric information: *we consider multiple scales of locality in a persistent framework for*
143 *local homology.* Concerning the overall construction, Stolz et al. (2020) is the closest to our method.
144 However, the authors assume that the intrinsic dimension is known and the proposed algorithm uses
145 a fixed scale, whereas our approach (i) operates in a multi-scale setting, (ii) provides local estimates
146 of intrinsic dimensionality of the data space, and (iii) incorporates model spaces that serve as a com-
147 parison. We can thus measure the deviation from an idealised manifold, requiring fewer assumptions
148 on the structure of the input data (Section 5.4 demonstrates the benefits of this perspective).

## 4 METHODS

150 Our framework TOAST (Topological Algorithm for Singularity Tracking) consists of two parts:
151 (i) a method to calculate a local intrinsic dimension of the data, and (ii) *Euclidicity*, a measure for
152 assessing the multi-scale deviation from a Euclidean space. TOAST is based on the assumption that
153 the intrinsic dimension of some given data is *not* necessarily constant across the data set, and is
154 best described by *local measurements*, i.e. measurements in a small neighbourhood of a given point.
155 Since there is no canonical choice for the magnitude of such a neighbourhood, TOAST is built on a
156 multi-scale analysis of data. Our main idea involves constructing a collection of local (punctured)
157 neighbourhoods for varying locality scales, and subsequently recording their topological features.
158 This procedure allows us to approximate local topological features (specifically, local homology)
159 of a given point, which we use to measure the intrinsic dimensionality of a space. Moreover, by
160 calculating the distance to Euclidean model spaces, we are capable of detecting singularities in a
161 large range of input data sets. Subsequently, we will briefly describe the 'moving parts' of TOAST;
162 please refer to Appendix A.1 for a terminology list.

### 4.1 PERSISTENT INTRINSIC DIMENSION

164 For a finite metric space $(\mathbb{X}, \mathrm{d})$ and $x \in \mathbb{X}$, let $B_r^s(x) := \{y \in \mathbb{X} \mid r \leq \mathrm{d}(x,y) \leq s\}$ denote
165 the intrinsic annulus of $x$ in $\mathbb{X}$ with respect to the parameters $r$ and $s$. Moreover, let $\mathcal{F}$ denote a
166 procedure that takes as input a finite metric space and outputs an ascending filtration of topological
167 spaces—such as a Vietoris–Rips filtration. By applying $\mathcal{F}$ to the intrinsic annulus of $x$, we obtain
168 a tri-filtration $(\mathcal{F}(B_r^s(x), t))_{r,s,t}$, where $t$ corresponds to the respective filtration step that is deter-
169 mined by $\mathcal{F}$. Note that this tri-filtration is covariant in $s$ and $t$, but contravariant in $r$; we denote it by
170 $\mathcal{F}(B_\bullet^\bullet(x), \bullet)$. Applying $i$th homology to this filtration yields a tri-parameter persistent module that
171 we call $i$th **persistent local homology (PLH)** of $x$, denoted by $\mathrm{PLH}_i(x; \mathcal{F}) := \mathrm{PH}_i(\mathcal{F}(B_\bullet^\bullet(x), \bullet))$.
172 To the best of our knowledge, this is the first time that PLH is considered as a multi-parameter per-
173 sistence module. Since the Vietoris–Rips filtration is the pre-eminent filtration in TDA, we will also
174 use the abbreviated notation $\mathrm{PLH}_i(x) := \mathrm{PLH}_i(x; \mathcal{V})$.

175 Our PLH formulation enjoys stability properties similar to the seminal stability theorem in persistent
176 homology (Cohen-Steiner et al., 2007), making it robust to small parameter changes (we assess
177 empirical stability in Section 5.1).

178 **Theorem 1.** *Given a finite metric space $\mathbb{X}$ and $x \in \mathbb{X}$, let $B_r^s(x)$ and $B_{r'}^{s'}(x)$ denote two*
179 *intrinsic annuli with $|r - r'| \leq \epsilon_1$ and $|s - s'| \leq \epsilon_2$. Furthermore, let $\mathcal{D}, \mathcal{D}'$ denote the*

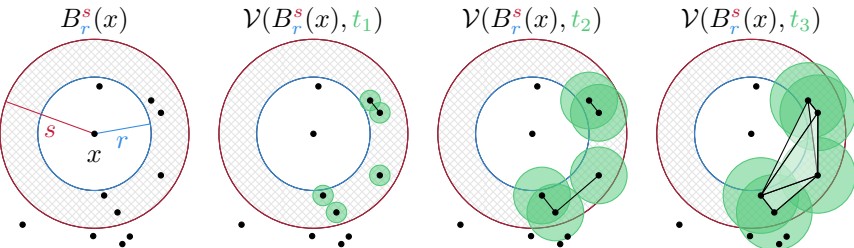

Figure 3: The intrinsic annulus $B_r^s(x)$ around a point $x$ in a metric space $(\mathbb{X}, \mathrm{d})$, as well as three filtration steps with varying $t$ parameters. By adjusting $r$ and $s$, we obtain a tri-filtration.

persistence diagrams corresponding to $\mathrm{PH}_i(B_r^s(x); \mathcal{V})$ and $\mathrm{PH}_i(B_{r'}^{s'}(x); \mathcal{V})$, respectively. Then $\frac{1}{2}\,\mathrm{d_B}(\mathcal{D}, \mathcal{D}') \leq \max\{\epsilon_1, \epsilon_2\}$.

For a finite set of points $\mathbb{X} \subset \mathbb{R}^N$ and $x \in \mathbb{X}$, we define the **persistent intrinsic dimension (PID)** of $x$ at scale $\epsilon$ as $i_x(\epsilon) := \max\{i \in \mathbb{N} \mid \mathrm{PH}_{i-1}(B_r^s(x)) \neq 0 \text{ for some } r \text{ and } s \text{ with } s < \epsilon\}$. This measure serves as a multi-scale characterisation of the intrinsic dimension of a data set. In case our data set constitutes a manifold sample, it turns out that we can recover the correct dimension.

**Theorem 2.** *Let $M \subset \mathbb{R}^N$ be an $n$-dimensional compact smooth manifold and let $\mathbb{X} := \{x_1, \ldots, x_S\}$ be a collection of uniform samples from $M$. For a sufficiently large $S$, there exist constants $\epsilon_1, \epsilon_2 > 0$ such that $i_x(\epsilon) = n$ for all $\epsilon_1 < \epsilon < \epsilon_2$ and any point $x \in \mathbb{X}$. Moreover, $\epsilon_1$ can be chosen arbitrarily small by increasing $S$.*

The implication of Theorem 2 is that $i_x(\epsilon)$ computes the correct intrinsic dimension of $M$ in a certain range of values $\epsilon > 0$, provided the sample is sufficiently large. In particular, $i_x(\epsilon)$ persists in this range, which suggests to consider a collection of $i_x(\epsilon)$ for varying $\epsilon$ to analyse the intrinsic dimension of $x$. We also have the following corollary, which specifically addresses stratified spaces (such as the 'pinched torus'), implying that our method can correctly detect the intrinsic dimension of individual strata. PID is thus capable of handling large classes of 'non-manifold' data sets.

**Corollary 1.** *Let $X = X_n \supset X_{n-1} \supset X_{n-2} \supset \cdots \supset X_{-1} = \emptyset$ be an $n$-dimensional compact stratified simplicial complex, s.t. $X_i \setminus X_{i-1}$ is smooth for every $i$. For a fixed $i$, let $\mathbb{X}_i := \{x_1, \ldots, x_S\}$ be a collection of uniform samples from $X_i \setminus X_{i-1}$. For a sufficiently large $S$, there are constants $\epsilon_1, \epsilon_2 > 0$ such that $i_x(\epsilon) = i$ for all $\epsilon_1 < \epsilon < \epsilon_2$ and any point $x \in \mathbb{X}_i$. Moreover, $\epsilon_1$ can be chosen arbitrarily small by increasing $S$.*

### 4.2 EUCLIDICITY

Knowledge about the intrinsic dimension of a neighbourhood is crucial for measuring to what extent such a neighbourhood deviates from being Euclidean. We refer to this deviation as *Euclidicity*, with the understanding that low values indicate Euclidean neighbourhoods while high values indicate singular regions of a data set. *Euclidicity* can be calculated without stringent assumptions on manifoldness: let $\mathbb{X} \subset \mathbb{R}^N$ be a finite data set, $x \in \mathbb{X}$ a point, and assume that we are given an estimate $n$ of the intrinsic dimension of $x$. In particular, the previously-described PID estimation procedure is applicable in this setting and may be used to obtain $n$, for example by calculating statistics on the set of $i_x(\epsilon)$ for varying locality parameters $\epsilon$. Euclidicity, however, can also make use of other dimensionality estimation procedures (see Camastra & Staiano (2016) for a survey). To assess how far a given neighbourhood of $x$ is from being Euclidean, we compare it to a Euclidean model space by measuring the deviation of their corresponding persistent local homology features. We start by defining the Euclidean annulus $\mathbb{E}B_r^s(x)$ of $x$ for parameters $r$ and $s$ to be a set of random uniform samples of $\{y \in \mathbb{R}^n \mid r \leq \mathrm{d}(x, y) \leq s\}$ such that $|\mathbb{E}B_r^s(x)| = |B_r^s(x)|$. Here, $r$ and $s$ correspond to the inner and outer radius of the Euclidean annulus, respectively. For $r' \leq r$ and $s \leq s'$ we extend $\mathbb{E}B_r^s(x)$ by sampling additional points to obtain $\mathbb{E}B_{r'}^{s'}(x)$ with $|\mathbb{E}B_{r'}^{s'}(x)| = |B_{r'}^{s'}(x)|$. Iterating this procedure leads to a tri-filtration $(\mathcal{F}(\mathbb{E}B_r^s(x), t))_{r,s,t}$ for any filtration $\mathcal{F}$, following our description in Section 4.1. We now define the persistent local homology of a Euclidean model space as

$$\mathrm{PLH}_i^{\mathbb{E}}(x; \mathcal{F}) := \mathrm{PH}_i(\mathcal{F}(\mathbb{E}B_\bullet^\bullet(x), \bullet)). \tag{2}$$

Again, for a Vietoris–Rips filtration $\mathcal{V}$, we use a short-form notation, i.e. $\mathrm{PLH}_i^{\mathbb{E}}(x) := \mathrm{PLH}_i^{\mathbb{E}}(x; \mathcal{V})$. Notice that $\mathrm{PLH}_i^{\mathbb{E}}(x)$ implicitly depends on the choice of intrinsic dimension $n$, and on the samples that are generated randomly. To remove the dependency on the samples, we consider $\mathrm{PLH}_i^{\mathbb{E}}(x)$ to be a sample of a random variable $\mathbf{PLH}_i^{\mathbb{E}}(x)$. Let $\mathrm{D}(\cdot, \cdot)$ be a distance measure for 3-parameter persistence modules, such as the *interleaving distance*.[4] We then define the **Euclidicity** of $x$, denoted by $\mathfrak{E}(x)$, as the expected value of these distances, i.e.

$$\mathfrak{E}(x) := \mathrm{E}\Big[\mathrm{D}\Big(\mathrm{PLH}_{n-1}(x), \mathbf{PLH}_{n-1}^{\mathbb{E}}(x)\Big)\Big]. \tag{3}$$

This quantity essentially assesses how far $x$ is from admitting a regular Euclidean neighbourhood.

**Implementation.** Calculating $\mathfrak{E}(x)$ requires different choices, namely (i) a range of locality scales, (ii) a filtration, and (iii) a distance metric between filtrations $\mathrm{D}$. Using a grid $\Gamma$ of possible radii $(r, s)$ with $r < s$, we approximate Eq. (3) using the *mean of the bottleneck distances of fibred Vietoris–Rips barcodes*, i.e.

$$\mathfrak{E}(x) \approx \mathrm{D}\big(\mathrm{PLH}_i(x), \mathrm{PLH}_i^{\mathbb{E}}(x)\big) := \frac{1}{C} \sum_{(r,s)\in\Gamma} \mathrm{d}_{\mathrm{B}}(\mathrm{PH}_i(\mathcal{V}(B_r^s(x), \bullet)), \mathrm{PH}_i(\mathcal{V}(\mathbb{E}B_r^s(x), \bullet))), \tag{4}$$

where $C$ is equal to the number of summands and $\mathrm{PLH}_i^{\mathbb{E}}(x)$ refers to a sample from a Euclidean annulus of the same size as the intrinsic annulus around $x$. Eq. (4) can be implemented using effective persistent homology calculation methods (Bauer, 2021), thus permitting an integration into existing TDA and machine learning frameworks (The GUDHI Project, 2015; Tauzin et al., 2020). Appendix A.4 provides pseudocode implementations, while Section 5 discusses how to pick these parameters in practice. We make one specific instantiation of our framework publicly available.[5]

**Properties.** The main appeal of our formulation is that calculating both PID and Euclidicity does not require strong assumptions about the input data. Treating dimension as a local quantity that is allowed to vary across multiple scales leads to beneficial expressivity properties. As we showed in Section 4.1, our method is *guaranteed* to yield the right values for manifolds and stratified simplicial complexes. This property substantially increases the practical applicability and expressivity, enabling our framework to handle unions of manifolds of varying dimensions, for instance. We require only a basic assumption, namely that the intrinsic dimension $n$ of the given data space is significantly lower than the ambient dimension $N$, making Euclidicity broadly applicable. Similar to curvature, Euclidicity makes use of the fact that one can compare data to 'model spaces,' allowing for different future adjustments.

**Limitations.** Our implementation of Euclidicity makes use of the Vietoris–Rips complex, which is known to grow exponentially with increasing dimensionality. While all calculations of Eq. (3) can be performed *in parallel*—thus substantially improving scalability vis-à-vis persistent homology on the complete input data set, both in terms of dimensions and in terms of samples—the memory requirements for a full Vietoris–Rips complex construction may still prevent our method to be applicable for certain high-dimensional data sets. This can be mitigated by selecting a different filtration (Anai et al., 2020; Sheehy, 2013); our proofs do not assume a specific filtration, and we leave the treatment of filtration-specific theoretical properties for future work. Finally, we remark that the reliability of the Euclidicity score depends on the validity of the intrinsic dimension; otherwise, the comparison does not take place with respect to the appropriate model space.

## 5 EXPERIMENTS

We demonstrate the expressivity of our proposed TOAST procedure in different settings, empirically showing that it (i) calculates the correct intrinsic dimension, and (ii) detects singularities when analysing data sets with known singular points. We also conduct a comparison with one-parameter approaches, showcasing how our multi-scale approach results in more stable outcomes. Finally, we analyse Euclidicity scores of benchmark datasets, giving evidence that our technique can be used as a measure for the geometric complexity of data.

---

[4] In our implementation, we will approximate this distance via the bottleneck distance.

[5] See the supplementary materials for the code and experiments.

## 5.1 PARAMETER SELECTION

Since Eq. (3) intrinsically incorporates multiple scales of locality, we need to specify an upper bound for the radii $(r_{\min}, r_{\max}, s_{\min}, s_{\max})$ that define the respective annuli in practice. Given a point $x$, we found the following procedure to be useful in practice: we set $s_{\max}$, i.e. the maximum of the outer radius, to the distance to the $k$th nearest neighbour of a point, and $r_{\min}$, i.e the minimum inner radius, to the smallest non-zero distance to a neighbour of $x$. Finally, we set the minimum outer radius $s_{\min}$ and the maximum inner radius $r_{\max}$ to the distance to the $\lfloor \frac{k}{3} \rfloor$th nearest neighbour. While we find $k = 50$ to yield sufficient results, spaces with a high intrinsic dimension may require larger values. The advantage of using such a parameter selection procedure is that it works in a data-driven manner, accounting for differences in density. Since our approach is inherently *local*, we need to find a balance between sample sizes that are sufficiently large to contain topological information, while at the same time being sufficiently small to retain a local perspective. We found the given range to be an appropriate choice in practice. As for the number of steps, we discretise the parameter range using 20 steps by default. Higher numbers are advisable when there are large discrepancies between the radii, for instance when $s_{\max} \gg r_{\max}$.

## 5.2 PERSISTENT INTRINSIC DIMENSION IS EXPRESSIVE

|  | METHOD | MIN | $\mu \pm \sigma$ | MAX |
|---|---|---|---|---|
| 1D | `lpca` | 1.00 | 1.42±0.78 | 3.00 |
|  | `twoNN` | 0.83 | 1.00±0.07 | 1.20 |
|  | `DANCo` | 1.00 | 1.00±0.01 | 1.16 |
|  | **PID** | 1.00 | 1.12±0.24 | 1.97 |
| 2D | `lpca` | 2.00 | 2.88±0.32 | 3.00 |
|  | `twoNN` | 1.01 | 1.90±0.36 | 2.53 |
|  | `DANCo` | 1.00 | 2.10±0.32 | 3.00 |
|  | **PID** | 1.52 | 1.95±0.06 | 2.08 |

Table 1: Dimensionality estimates for the concatenation of $S^1$ and $S^2$.

We first analyse the behaviour of persistent intrinsic dimension (PID) on samples from a space obtained by concatenating $S^1$ (a circle) and $S^2$ (a sphere) at a gluing point. Table 1 shows a comparison of PID with state-of-the-art dimensionality estimators.[6] We find that PID outperforms all estimators in terms of mean and standard deviation for the 2D points, thus correctly indicating that the majority of all points admit non-singular 2D neighbourhoods. For the 1D points, the mean of the dimensionality estimate of PID is still close to the ground truth, while its standard deviation and maximum correctly capture the fact that some 1D points are situated closer to the gluing point. This behaviour is in line with our philosophy of considering dimensionality as an inherently local phenomenon. In case such behaviour is not desirable for a specific data set, Euclidicity calculations support *any* dimensionality estimator; since such estimators do not come with strong guarantees such as Theorem 2, their choice must be ultimately driven by the data set at hand. See Appendix A.6 for a more detailed analysis of these estimates.

**Stability.** In practice, the sample density may not be sufficiently high for Theorem 2 to apply. This means that there may appear artefact homological features in dimensions *higher* than the intrinsic dimension of a given space. We thus only consider features that exceed a certain persistence threshold in comparison to the persistence of features of lower dimension: for any data point $x$ and the respective intrinsic annulus $B_r^s(x)$, we eliminate all topological features whose lifetimes are smaller than the maximum lifetime of features in one dimension below. This results in markedly stable estimates of intrinsic dimension, which are less prone to overestimations.

## 5.3 EUCLIDICITY CAPTURES SINGULARITIES

Fig. 1 shows that Euclidicity is capable of detecting the singularity of the 'pinched torus.' Of particular relevance is the fact that Euclidicity also highlights that points in the vicinity of the singular point are *not* fully regular. This is an important property for practical applications since it implies that Euclidicity can detect such *isolated singularities* even in the presence of sampling errors.

Besides the pinched torus, another prototypical example of singular spaces is given by $S^n \vee S^n$, the wedge of two $n$-dimensional spheres. Intuitively, $S^n \vee S^n$ is obtained by two $n$-dimensional spheres that are glued together at a certain point. Denoting the gluing point by $x_0$, for a suitable triangulation of $X = S^n \vee S^n$, this space is naturally stratified by $X \supset \{x_0\}$. Next, we apply TOAST to samples

---

[6]Method names are taken from the `scikit-dimension` toolkit. See Appendix A.6 for more details.

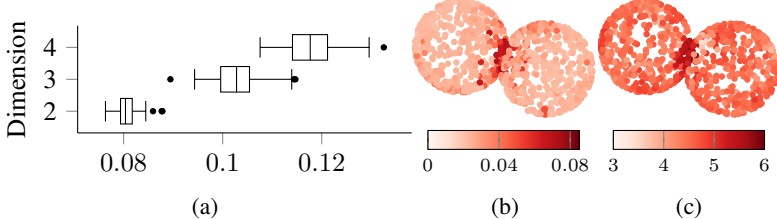

Figure 4: (a): Euclidicity scores of wedged spheres for different dimensions. High values indicate singular points/neighbourhoods. The Euclidicity of the singular point always constitutes a clear positive outlier. In 2D, *Euclidicity* (b) results in a clearly-delineated singular region when compared to a single-parameter score (c).

of such wedged spheres of dimensions $2, 3$ and $4$, calculating their respective Euclidicity scores. Since larger intrinsic dimensions require higher sample sizes to maintain the same density, we start with a sample size of 20000 in dimension 2 and increase it consecutively by a factor of 10. We then calculate Euclidicity of 50 random samples in the respective data set, and additionally for the singular point $x_0$. Fig. 4a shows the results of our experiments. We observe that the singular point possesses a *significantly higher Euclidicity* score than the random samples. Moreover, we find that Euclidicity scores of non-singular points exhibit a high degree of variance across the data, which is caused by the fact that the sampled data does not perfectly fit the underlying space the points are being sampled from. This strengthens our main argument: assessing whether a specific point is Euclidean does not require a binary decision but a continuous measure such as Euclidicity.

**Stability.** As predicted by Theorem 1, Euclidicity estimates are stable in practice. We first note that Euclidicity is *robust towards sampling*: repeating the calculations for the 'pinched torus' over different batches results in highly similar distributions that are not distinguishable according to Tukey's range test (Tukey, 1949) at the $\alpha = 0.05$ confidence level. Moreover, choosing larger locality scales still enables us to detect singularities at higher computational costs and incorporating larger parts of the point cloud. Please refer to Appendix A.5 for a more detailed discussion of this aspect.

## 5.4 EUCLIDICITY IS MORE EXPRESSIVE THAN SINGLE-PARAMETER APPROACHES

Our Euclidicity measure leads to significantly more stable results than a comparable one-parameter approach for geometry-based anomaly detection (Stolz et al., 2020): Fig. 4b and Fig. 4c compare multi-parameter Euclidicity with one-parameter Euclidicity for 20000 samples of $S^2 \vee S^2$. The constant-scale approach results in many points with high anomaly scores that in fact *do* admit a Euclidean neighbourhood. We quantify this by analysing the empirical distributions of anomaly scores of the two data spaces (see Appendix A.8 for more details), with the one-parameter method exhibiting a much larger variance than our multi-parameter Euclidicity measure. The multi-parameter distribution shows that the mass is concentrated around the mean, but also contains outliers with high Euclidicity scores. These outliers correspond to points in the data space whose distance to the singular point is small. We thus conclude that Euclidicity scores increase once one approaches the singularity—which is *not* the case for single-parameter methods with a fixed locality scale. In fact, the main advantage of Euclidicity is that it implicitly incorporates information about the scale on which a given data point admits a Euclidean neighbourhood.

## 5.5 EUCLIDICITY CAPTURES GEOMETRIC COMPLEXITY OF HIGH-DIMENSIONAL SPACES

To test TOAST in an unsupervised setting, we calculate Euclidicity scores for the MNIST and FASHIONMNIST data sets, selecting mini-batches of 1000 samples from a subsample of 10000 random images of these data sets. Following Pope et al. (2021), we assume an intrinsic dimension of 10; moreover, we use $k = 50$ neighbours for local scale estimation. To ensure that our results are representative, we repeat all calculations for five different subsamples. Euclidicity scores range from $[1.1, 5.3]$ for MNIST, and $[1.3, 5.6]$ for FASHIONMNIST. The scores of the two datasets appear to be following different distributions (see Appendix A.7 for a visualisation and a more detailed depiction of the distributions).

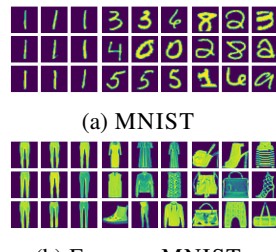

(a) MNIST

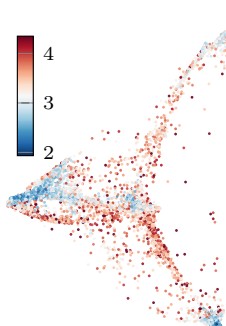

(b) FASHIONMNIST

Figure 5: Left to right: low, median, high Euclidicity.

Fig. 5 shows a selection of 9 images, corresponding to the lowest, median, and highest Euclidicity scores, respectively. We observe that high Euclidicity scores correspond to images with a high degree of non-linearity, whereas low Euclidicity scores correspond to images that exhibit less complex structures: for MNIST, these are digits of '1.' Interestingly, we observe the same phenomenon for FASHION-MNIST, where images with low Euclidicity ('pants') possess less geometric complexity in contrast to images with high Euclidicity. Since low Euclidicity can also be seen as an indicator of how close a neighbourhood is to being *locally linear*, this finding hints at the existence of simple substructures in such data sets. Euclidicity could thus be used as an unsupervised measure of geometric complexity.

### 5.6 EUCLIDICITY CAPTURES LOWER-DIMENSIONAL STRUCTURES IN CYTOMETRY DATA

To highlight the utility of Euclidicity in unsupervised representation learning, we calculate it on an induced pluripotent stem cell (iPSC) reprogramming data set (Zunder et al., 2015). The data set depicts a progression of so-called fibroblasts diverging, and splitting into two different lineages. Fig. 6 shows an embedding obtained via PHATE (Moon et al., 2019) and the Euclidicity scores of the original data. We find that high Euclidicity scores correspond to points that exhibit a *lower density* in the embedding, being in fact situated in lower-dimensional subspaces. Since lower-dimensional points in a space can be considered *singular* in the sense of stratified spaces, this is further evidence for Euclidicity to be a useful tool for detecting non-manifold regions in data. Please refer to Appendix A.9 for more details.

Figure 6: An embedding of the iPSC data with colours based on Euclidicity highlights dense non-singular regions.

## 6 DISCUSSION

We presented TOAST, a novel framework for locally estimating the intrinsic dimension (via PID, the persistent intrinsic dimension) and the 'manifold-ness' (via Euclidicity, a multi-scale measure of the deviation from Euclidean space) of point clouds. Our method is based on a novel formulation of persistent local homology as a multi-parameter approach, and we provide theoretical guarantees for it in a dense sample setting. Our experiments showed significant improvements of stability compared to geometry-based anomaly detection methods with fixed locality scales, and we found that Euclidicity can detect singular regions in data sets with known singularities. Using high-dimensional benchmark data sets, we also observed that Euclidicity can serve as an *unsupervised measure of geometric complexity*.

For future work, we envision two relevant research directions. First and foremost will be the inclusion of Euclidicity into machine learning models to make them 'singularity-aware.' In light of our experiments in Section 5.5, we believe that Euclidicity could be particularly useful in unsupervised scenarios, or provide an additional weight in classification settings (to ensure that singular examples are being given lower confidence scores). Moreover, Euclidicity could be used in the detection of adversarial samples—a task for which knowledge about the underlying topology of a space is known to be crucial (Jang et al., 2020). As a second direction, we want to further improve the properties of Euclidicity itself. To this end, we plan to investigate if incorporating custom distance measures for three-parameter persistence modules, i.e. different metrics for Eq. (4), lead to improved results in terms of stability, expressivity, or computational efficiency. Moreover, we hypothesise that replacing the Vietoris–Rips filtration by other constructions (de Silva & Carlsson, 2004) could prove beneficial in reducing the number of samples for calculating Euclidicity. Along these lines, we also plan to derive theoretical results that relate specific filtrations and the expressivity of the corresponding Euclidicity measure. Another direction for future research concerns the approximation of a manifold from inherently singular data, i.e. finding the *best* manifold approximation to a given data set with singularities. This way, singularities could be resolved during the training phase of models, provided an appropriate loss function exists. Euclidicity may thus serve as a metric for assessing data sets, paving the way towards more trustworthy and faithful embeddings.

## REPRODUCIBILITY STATEMENT

We provide our code as part of the supplementary materials. All dependencies are listed in the respective `pyproject.toml` file, and the `README` discusses how to install our package and run our experiments. Our implementation leverages multiple CPUs if available but has no specific hardware requirements otherwise.

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

# A APPENDIX

## A.1 NOTATION

| Symbol | Meaning |
|---|---|
| $\epsilon$ | local annulus scale parameter |
| $\mathbb{R}$ | real numbers |
| $H_i$ | $i$th (ordinary) homology functor (with $\mathbb{Z}/2\mathbb{Z}$ coefficients) |
| $\tilde{H}_i$ | $i$th reduced homology functor (with $\mathbb{Z}/2\mathbb{Z}$ coefficients) |
| $\inf$ | infimum |
| $\sup$ | supremum |
| $|\cdot|_\infty$ | uniform (infinity) norm |
| $n$ | intrinsic dimension of the space under consideration |
| $N$ | ambient dimension of the space under consideration |
| $\varinjlim$ | (categorical) colimit |
| $S^k$ | $k$-dimensional sphere |
| $c^\circ X := X \times (0,1]/X \times \{1\}$ | open cone of a topological space $X$ |

## A.2 PROOFS OF THE MAIN STATEMENTS IN THE PAPER

We restate the theorems from the main paper for the convenience of readers, along with their proofs, which were removed for space reasons. We first prove the stability theorem, first stated on p. 5 in the main text, which shows that our method enjoys stability properties with respect to radius changes of the intrinsic annuli.

**Theorem 1.** *Given a finite metric space* $\mathbb{X}$ *and* $x \in \mathbb{X}$, *let* $B_r^s(x)$ *and* $B_{r'}^{s'}(x)$ *denote two intrinsic annuli with* $|r - r'| \le \epsilon_1$ *and* $|s - s'| \le \epsilon_2$. *Furthermore, let* $\mathcal{D}, \mathcal{D}'$ *denote the persistence diagrams corresponding to* $\mathrm{PH}_i(B_r^s(x); \mathcal{V})$ *and* $\mathrm{PH}_i(B_{r'}^{s'}(x); \mathcal{V})$, *respectively. Then* $\frac{1}{2} \mathrm{d}_\mathrm{B}(\mathcal{D}, \mathcal{D}') \le \max\{\epsilon_1, \epsilon_2\}$.

*Proof.* The Hausdorff distance of two non-empty subsets $A, B \subset \mathbb{X}$ is $\mathrm{d}_\mathrm{H}(A, B) := \inf\{\epsilon \ge 0 \mid A \subset B_\epsilon, B \subset A_\epsilon\}$, where $A_\epsilon = \cup_{a \in A}\{x \in \mathbb{X}; \mathrm{d}(x, a) \le \epsilon\}$ denotes the $\epsilon$-thickening of $A$ in $X$. Set $\epsilon := \max\{\epsilon_1, \epsilon_2\}$. By assumption, $B_r^s(x) \subset B_{r'}^{s'}(x)_\epsilon$ and $B_{r'}^{s'}(x) \subset B_r^s(x)_\epsilon$, i.e. $\mathrm{d}_\mathrm{H}(B_r^s(x), B_{r'}^{s'}(x)) \le \epsilon$. Using the geometric stability theorem of persistence diagrams (Chazal et al., 2014), we have $\frac{1}{2} \mathrm{d}_\mathrm{B}(\mathcal{D}, \mathcal{D}') \le \mathrm{d}_\mathrm{H}(B_r^s(x), B_{r'}^{s'}(x))$, which proves the claim. $\square$

Next, we prove that our *persistent intrinsic dimension* (PID) measure is capable of capturing the dimension of manifolds correctly, provided sufficiently many samples are present. This theorem was first stated on p. 5.

**Theorem 2.** *Let* $M \subset \mathbb{R}^N$ *be an* $n$-dimensional compact smooth manifold and let $\mathbb{X} := \{x_1, \ldots, x_S\}$ *be a collection of uniform samples from* $M$. *For a sufficiently large* $S$, *there exist constants* $\epsilon_1, \epsilon_2 > 0$ *such that* $i_x(\epsilon) = n$ *for all* $\epsilon_1 < \epsilon < \epsilon_2$ *and any point* $x \in \mathbb{X}$. *Moreover,* $\epsilon_1$ *can be chosen arbitrarily small by increasing* $S$.

*Proof.* Let $x \in \mathbb{X}$ be an arbitrary point. Since $M$ is a manifold, $x$ admits a Euclidean neighbourhood $U$. Since $M$ is smooth, we can assume $U$ to be arbitrarily close to being flat by shrinking it. Thus, we can find $\epsilon_2 > 0$ with $B_r^s(x) \subset U$ for all $r, s < \epsilon_2$ such that $\mathrm{H}_i(\mathcal{V}(B_r^s(x), t)) = 0$ for all $i \ge n$, and all $t$. Hence, $\mathrm{PH}_i(B_r^s(x)) = 0$ for all $i \ge n$, and therefore $i_x(\epsilon_2) \le n$. By contrast, for $S$ sufficiently large, and $r, s$ as before, there exists a parameter $t$ such that $\mathcal{V}(B_r^s(x), t)$ is homotopy-equivalent to an $(n-1)$-sphere, and so $\mathrm{H}_{n-1}(\mathcal{V}(B_r^s(x), t))$ admits a generator, i.e. it is non-trivial. Consequently, $\mathrm{PH}_{n-1}(B_r^s(x)) \ne 0$, and $i_x(\epsilon_2) = n$. By further increasing $S$, we can ensure that the statement still holds when we decrease $\epsilon_2$, which proves the two remaining claims. $\square$

### A.3 ADDITIONAL PROOFS

To make this paper self-contained, we provide a brief proof of Eq. (1). By the excision axiom for homology, we have

$$H_i(X, X \setminus x) \cong H_i(St(x), St(x) \setminus x). \tag{5}$$

Since $St(x)$ is *contractible*, the long exact reduced homology sequence of the pair $(St(x), St(x) \setminus x)$ records exactness of

$$0 = \tilde{H}_i(St(x)) \to H_i(St(x), St(x) \setminus x) \to \tilde{H}_{i-1}(St(x) \setminus x) \to \tilde{H}_{i-1}(St(x)) = 0$$

for all $i$, and therefore $H_i(St(x), St(x) \setminus x) \cong \tilde{H}_{i-1}(St(x) \setminus x)$. Eq. (1) now follows from the observation that $St(x) \setminus x$ deformation retracts to $Lk(x)$.

### A.4 PSEUDOCODE

We provide brief pseudocode implementations of the algorithms discussed in Section 4. In the following, we use $\# Bar_i(\mathbb{X})$ to denote the number of $i$-dimensional persistent barcodes of $\mathbb{X}$ (w.r.t. the Vietoris–Rips filtration, but any other choice of filtration affords the same description). Algorithm 1 explains the calculation of *persistent intrinsic dimension* (see Section 4.1 in the main paper for details). For the subsequent algorithms, we assume that the estimated dimension of the intrinsic dimension of the data is $n$. We impose no additional requirements on this number; it can, in fact, be obtained by any choice of intrinsic dimension estimation method. As a short-hand notation, for $p_i = PH_{n-1}(\mathcal{V}(\mathbb{EB}_\bullet^\bullet(x), \bullet))$ w.r.t. some sample of $\{y \in \mathbb{R}^n \mid r \leq d(x, y) \leq s\}$, we denote by $p_i^{r,s} = PH_{n-1}(\mathcal{V}(\mathbb{EB}_r^s(x), \bullet))$ the respective fibred persistent local homology barcode (calculated w.r.t. the same sample). Algorithm 2 then shows how to calculate the *Euclidicity* values, following Eq. (3) and one of its potential implementations, given in Eq. (4).

---

**Algorithm 1** An algorithm for calculating the *persistent intrinsic dimension* (PID)

---

**Require:** $x \in \mathbb{X}$, $s_{\max}$, $\ell$.

1: **for** $s \in \Gamma$ **do**                  ▷ Iterate over the parameter grid
2:      $i_x(s) \leftarrow 0$
3:      **for** $r < s \in \Gamma$ **do**
4:          **for** $i = 1, \dots, N - 1$ **do**
5:              **Calculate** $\# Bar_i(B_r^s(x))$
6:              **if** $\# Bar_i(B_r^s(x)) > 0$ **then**
7:                  $i_x(s) \leftarrow i + 1$
8:              **end if**
9:          **end for**
10:      **end for**
11:      **return** $i_x(s)$
12: **end for**

---

**Algorithm 2** An algorithm for calculating the *Euclidicity values* $\delta_{jk}$

---

**Require:** $x \in \mathbb{X}$, $s_{\max}$, $\ell$, $n$, $\{p_1, \dots, p_m\}$.

1: **for** $j = 1, \dots, m$ **do**
2:      **for** $k = j + 1, \dots, m$ **do**
3:          **for** $s \in \Gamma$ **do**
4:              **for** $r \in \Gamma, r < s$ **do**
5:                  **Calculate** $d_B(p_j^{r,s}, p_k^{r,s})$          ▷ Calculate bottleneck distance
6:                  **return** $d_B(p_j^{r,s}, p_k^{r,s})$
7:              **end for**
8:          **end for**
9:      **Calculate** $D(p_j, p_k)$                         ▷ Evaluate Eq. (4)
10:      **return** $D(p_j, p_k)$
11:      **end for**
12: **end for**

---

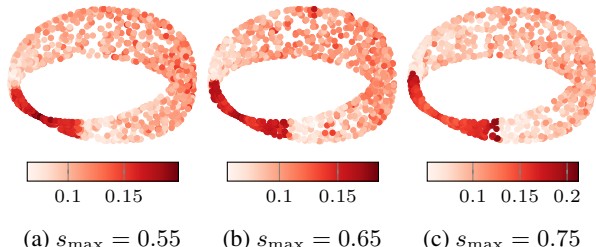

(a) $s_{\max} = 0.55$  (b) $s_{\max} = 0.65$  (c) $s_{\max} = 0.75$

Figure 8: Modifying the outer radius $s_{\max}$ still enables us to detect the singularity of the 'pinched torus.' Larger radii, however, progressively increase the field of influence of our method, thus starting to assign high Euclidicity values to larger regions of the point cloud.

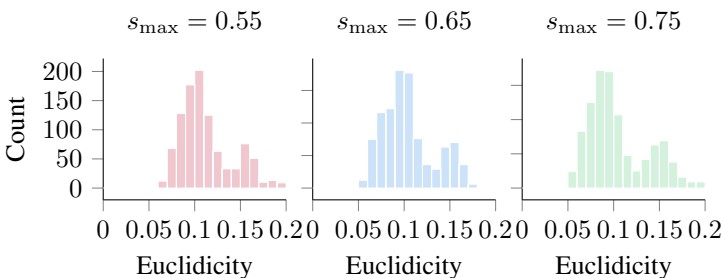

Figure 9: Histograms of the Euclidicity values for the point clouds shown in Fig. 8. Larger radii result in the distribution accumulating more probability mass at higher Euclidicity values, making the singularity detection procedure less local (but still succeeding in detecting the singularity and its environs).

## A.5  STABILITY OF EUCLIDICITY ESTIMATES

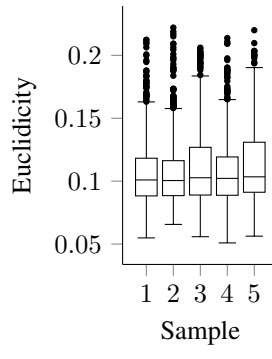

Figure 7: Boxplots of the Euclidicity values of different random samples of the 'pinched torus' data set. While each sample invariably exhibits some degree of geometric variation, we are able to reliably identify the singularity and its neighbourhood.

Fig. 7 shows that Euclidicity is robust under sampling; repeating the calculations for smaller batches of the 'pinched torus' data set (500 points each) still lets us detect the singularity and its neighbours reliably. This robustness is an important property in practice where we are dealing with samples from an unknown data set whose shape properties we want to capture. Euclidicity enables us to perform these calculations in a robust manner. Following the brief discussion in Section 5.1, we show the results of varying $s_{\max}$, the outer radius of the local annulus, for the 'pinched torus' data set. Fig. 8 depicts point clouds of 1000 samples; we observe that the singularity, i.e. the 'pinch point,' is always detected. For larger radii, however, this detection becomes progressively more *global*, incorporating larger parts of the point cloud. Fig. 9 depicts the corresponding histograms; we observe the same shift in probability mass towards the tail end of the distribution. For extremely large annuli, we estimate that we lose a clear distinction between singular values and non-singular values. Our data-driven parameter selection procedure is thus to be preferred in practice since it incorporates data density.

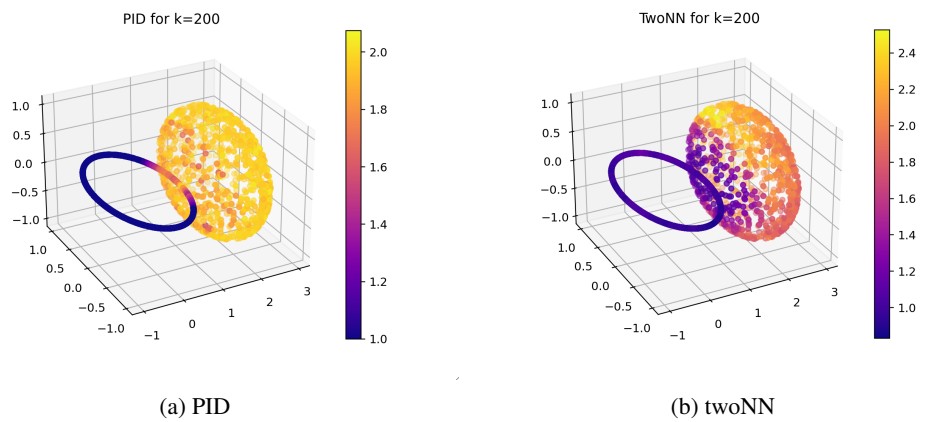

(a) PID  (b) twoNN

Figure 10: Even for large values of $k$, PID still does not overestimate the local dimensionality of the data, exhibiting a clear distinction between the circle and the sphere, respectively.

### A.6 COMPARISON OF PID WITH OTHER DIMENSION ESTIMATES

In order to assess the quality of PID, we decided to test its performance on a space that is both singular and has non-constant dimension. The data space we chose consists of 2000 samples of $S^1 \vee S^2$, i.e. a 1-sphere glued together with a 2-sphere at a certain concatenation point. We then applied the PID procedure for a maximum locality scale that was given by the $k$ nearest neighbour distances, for $k \in \{25, 50, 75, 100, 125, 150, 175, 200\}$. We assigned to each point the average of the PID scores at the respective scales that are less than or equal to the $k$ nearest neighbour bound. Subsequently, we compared the results with other local dimension estimates for the respective number of neighbours. The methods that were chosen for comparison include `lpca`, `twoNN`, `KNN`, and `DANCo`; we used the respective implementation from the `scikit-dimension` Python package.[7].

Fig. 10a shows the PID results for a maximum locality scale of 200 neighbours, with colours showing the estimated dimension values for each point. Overall, the correct intrinsic dimension is detected for most of the points. However, points that lie close to the singular point show a PID value between 1 and 2. Similarly to what we already discussed for Euclidicity, PID should therefore also be interpreted as a measure that incorporates the intrinsic dimension of a point on *multiple scales* of locality. For real-world data, the dimension will generally change when changing the locality scale. However, since there is no canonical choice of scale, we believe that any such scale provides valuable information about the intrinsic dimension that is worth being measured. We therefore argue that a multi-scale approach like ours is appropriate in practice, especially in a regime that is agnostic with respect to the underlying intrinsic dimension. By contrast, Fig. 10b shows the corresponding dimension estimates for twoNN, where we observe less stable and reliable results across the dataset.

Fig. 11a shows boxplots of the distributions of the dimension estimates, for all points that lie on the 1D-sphere. We see that for PID, the mass is concentrated at a value of 1. Although there are outliers present, these correspond to points that are close to the singularity, as it was expected. We note that other methods like `KNN` and `lpca` might highly overestimate the dimension, and that the interquartile range is significantly higher for `twoNN` and `KNN`. Fig. 11b shows the same distributions for the points that lie on the 2D-sphere. Again, `lpca` highly overestimates the dimension since the median lies at a value of 3. Again, the interquartile range of PID is the tightest, and the estimates are closest to the ground truth. Moreover, the lower-value outliers again correspond to points that are close to the singular gluing point.

Fig. 12a and Fig. 12b show average dimension estimate scores of all investigated methods for varying values of $k$, both for points on the 1-sphere and the 2-sphere. We note that on average, only `twoNN` and `DANCo` lead to results which are comparable with the reliability of our method. However, as we already saw in Fig. 11a and Fig. 11b, the variance of the scores of our method is significantly lower, leading to more reliable outputs for each of the points.

---

[7]https://scikit-dimension.readthedocs.io/en/latest/

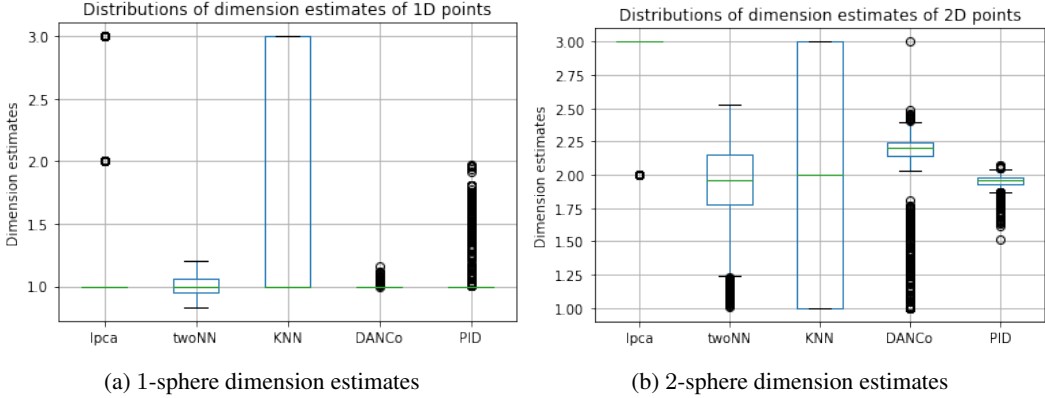

(a) 1-sphere dimension estimates        (b) 2-sphere dimension estimates

Figure 11: Estimates of the local intrinsic dimension for points that are close to the 1D-sphere, i.e. the circle, or the 2D-sphere, respectively.

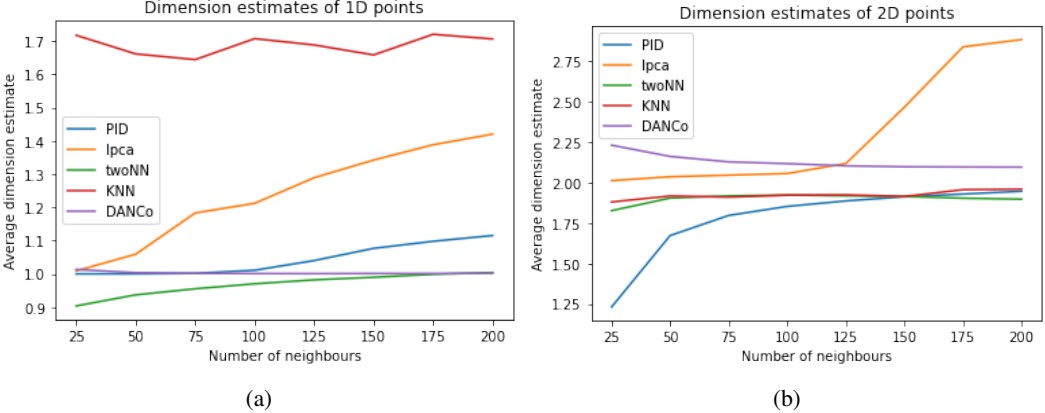

(a)                      (b)

Figure 12: Dimension estimates of the 1D-sphere and the 2D-sphere for different methods, plotted as a function of the number of neighours $k$.

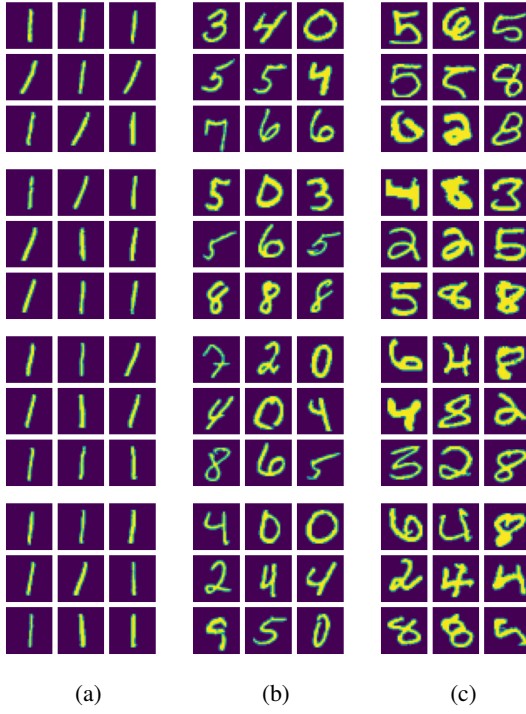

(a)        (b)        (c)

Figure 13: From left to right: more examples of low Euclidicity values, median Euclidicity values, and high Euclidicity values for the MNIST data set.

## A.7    EUCLIDICITY OF MNIST AND FASHIONMNIST

Fig. 13 and Fig. 14 show the Euclidicity results for the 4 additional runs on both the MNIST and FASHIONMNIST data sets. Again, we depicted the 9 images with lowest (left), medium (middle), and highest (right) Euclidicity scores for the two datasets. Moving from left to right, the images exhibit increases in the complexity of the local geometry, giving evidence for the reproducibility of the observation we remarked in Section 5.5.

Finally, as Fig. 15 shows, the empirical distributions of the calculated Euclidicity scores differ significantly for the MNIST and FASHIONMNIST data sets, with the distribution for MNIST exhibiting a bimodal behaviour, whereas the FASHIONMNIST Euclidicity value distribution is unimodal. We hypothesise that this corresponds to regions of simple complexity—and locally linear structures—in the MNIST data set, which are absent in the FASHIONMNIST data set.

## A.8    ONE-PARAMETER VERSUS MULTI-PARAMETER EUCLIDICITY FOR WEDGED SPHERES

Fig. 16 shows the empirical distributions of Euclidicity scores for fixed locality parameters (left) and for our proposed multi-scale locality approach (right). We see that the variance is *significantly lower* in the multi-scale regime, indicating more stable and robust results. Moreover, the ratio of maximum and mean is higher in the multi-parameter setting, where high Euclidicity scores correspond to data points that lie close to the singularity, resulting in more reliable outcomes.

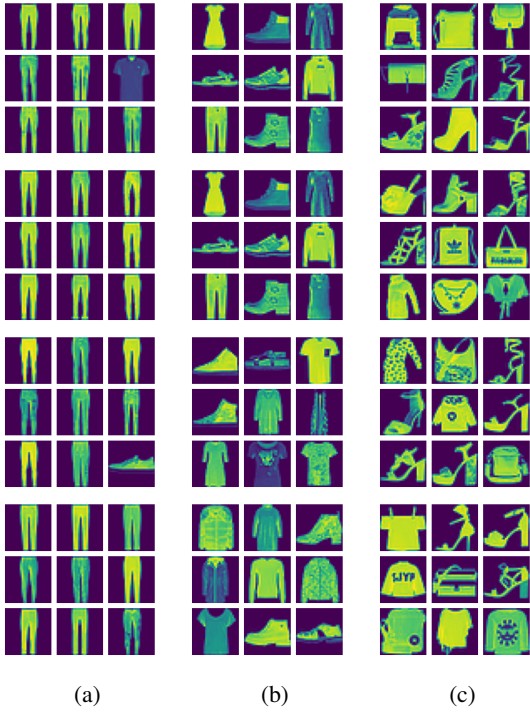

(a)      (b)      (c)

Figure 14: From left to right: more examples of low Euclidicity values, median Euclidicity values, and high Euclidicity values for the FASHIONMNIST data set.

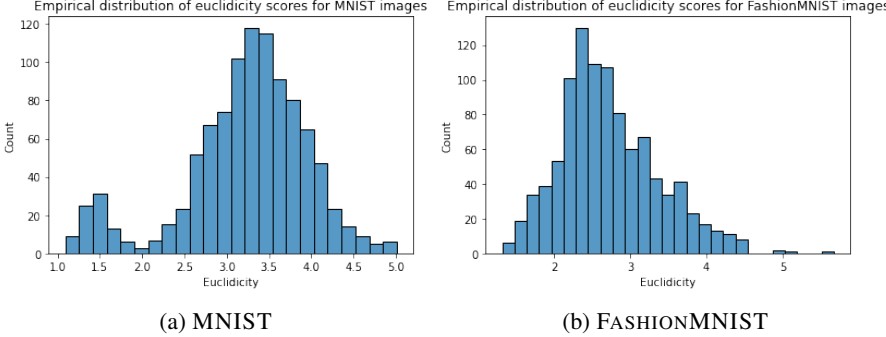

(a) MNIST            (b) FASHIONMNIST

Figure 15: Both MNIST and FASHIONMNIST exhibit markedly different distributions in terms of Euclidicity: MNIST Euclidicity values are bimodal, whereas FASHIONMNIST Euclidicity values are unimodal.

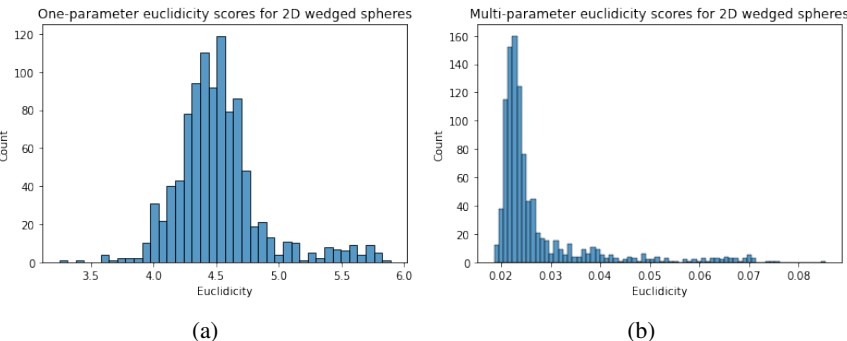

(a)  (b)

Figure 16: A comparison of Euclidicity values of a one-parameter approach (left) and our multi-parameter approach (right) demonstrates that multiple scales are necessary to adequately capture singularities.

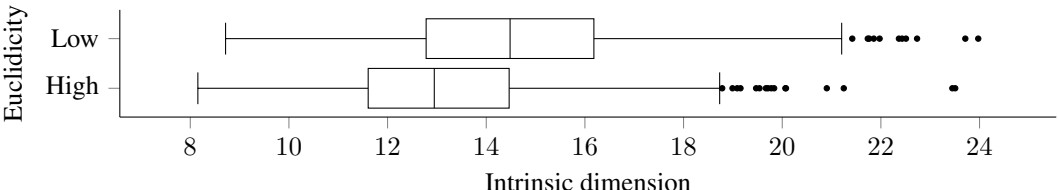

Figure 17: A comparison of intrinsic dimension estimates computed for points in the iPSC dataset that admit high (left) and low (right) Euclidicity scores. The `twoNN` dimensionality estimator was used for this example.

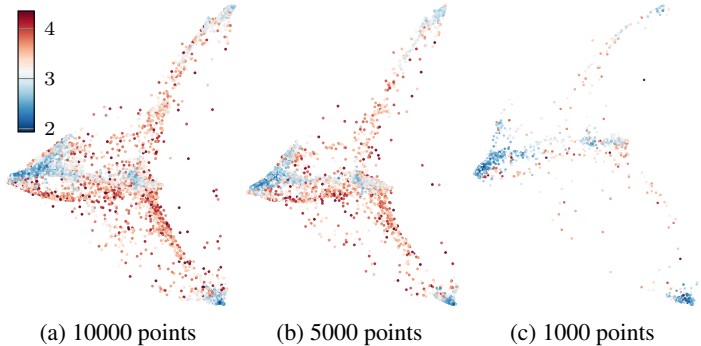

(a) 10000 points          (b) 5000 points          (c) 1000 points

Figure 18: Euclidicity remains stable under subsampling the iPSC data set. Minor variations in the point cloud shape are due to the PHATE embedding algorithm; Euclidicity was calculated on the raw data.

## A.9 EUCLIDICITY OF IPSC DATA

The iPSC data set Zunder et al. (2015) consists of 33 variables and around 220k samples. It is known to contain branching structures that can best be extracted using PHATE (Moon et al., 2019), a non-linear dimensionality reduction algorithm. We only employ this algorithm for *visualisation purposes*; all Euclidicity calculations are performed on the original data. Using `twoNN` for dimensionality estimation, we obtained a mean intrinsic dimension of 16; as outlined above, other dimensionality estimators may be employed as well—we consider this analysis to be a proof of concept first and foremost. We selected parameters as described in Section 5.5, and computed Euclidicity for 10000 samples.

We observe that high-Euclidicity scores correspond to points that exhibit a *lower density* in the PHATE embedding,[8] and according to the `twoNN` estimates we see that such points are in fact of *lower intrinsic dimension*; see Fig. 17 for details. More specifically, we calculated the intrinsic dimension for the subsample, observing that the interquartile range for the 1000 points with *highest Euclidicity* is around 12–14, whereas the interquartile range of the 1000 *lowest Euclidicity* points ranges between around 13–16. Again, we used the `twoNN` algorithm for intrinsic dimensionality estimates (using $k = 50$ nearest neighbours). Since lower-dimensional points in a space can be regarded as being singular in the sense of stratified spaces, we see further evidence for Euclidicity as a useful tool for the detection of non-manifold regions in the data. Finally, we remark that our analyses remain valid under *subsampling*. Fig. 18 depicts subsamples of different sizes for which we calculated Euclidicity (on the raw data, respectively, using PHATE to obtain embeddings). Euclidicity distributions remain stable and the same phenomena are highlighted for each subsample.

---

[8]However, notice that low-density regions in the PHATE visualisation need not necessarily correspond to low-density regions in the original dataset.

