# OpenReview forum: "TOAST: Topological Algorithm for Singularity Tracking"
_ICLR.cc/2023/Conference — Submitted to ICLR 2023_

### Official Review · Reviewer_vmhb · 2022-10-23

**Confidence:** 2
**Correctness:** 4
**Technical Novelty And Significance:** 3
**Empirical Novelty And Significance:** 3
**Recommendation:** 6

**Clarity, Quality, Novelty And Reproducibility:**

The proposed approach constitutes a significant contribution in the field, as long as it performs as expected. As regards the clarity, in general, I find papers related to topology a bit hard to access due to the terminology that is used and the background knowledge that I assumed. I think that the authors tried to give the high level idea in several places, but I believe that a suitable description (not too technical and not too abstract) of the approach is probably missing.

**Strength And Weaknesses:**

I think that the problem the paper aims to solve is very interesting and useful for modern machine learning, and it has been motivated well. I am not an expert in the field of topological data analysis, but I believe that the proposed approach differs from the previous related works which are properly presented. The technical content seems to be reasonable and to support the claims, however, I have not checked it thoroughly.

**Summary Of The Paper:**

The authors propose a technique based on topological data analysis to: 1. estimate pointwise the intrinsic dimensionality of the data manifold in the local neighbourhood, and 2. assess if the local neighbourhood resembles a Euclidean space with the same dimensionality. In this way, singularities on the data manifold can be found. The method has a multi-scale property as it takes into account neighbourhoods of different size. In the experiments it is demonstrated the effectiveness of the approach.

**Summary Of The Review:**

Questions:
1. I think that a brief summary of the approach is missing, while in several places (e.g. "Our contributions") you summarize from a high level point of view the steps of the approach. Perhaps, you could explain in a non-technical way each step e.g. what we need to do intuitively to estimate the intrinsic dimension of the local neighbourhood around a point.
2. I believe that you could save some space e.g. by moving proofs in the appendix, such that to include some additional figures to demonstrate the steps of the approach.
3. In high dimensions a lot of data are necessary in order to approximate the data manifold. I appreciate the synthetic and the MNIST/FashionMNIST experiments, but I think that some additional real world data in high dimensions should be conducted.

I think this is an interesting contribution with potential merits, buy as a non-expert in the field I cannot criticize objectively the actual contribution of the method. However, I believe that there is some room for improvement as regards the clarity and the accessibility.

---

> ### Author Response · Authors · 2022-11-08
> **Our initial response to your comments**
>
> Thank you very much for your review and the related suggestions. We are
> glad to have captured your interest in our project, and we are grateful
> for appreciating the **novelty** of the given approach, as well as the
> overall **quality** of the method. We believe that the suggestions for
> improvement provided by your review can easily be accommodated within
> the revision cycle, and we are currently preparing a new version of the
> manuscript.
>
> ## Clarity
>
> > I think that a brief summary of the approach is missing
>
> Thanks a lot for this suggestion! We are currently revising the
> manuscript to include a more high-level description in the manuscript.
> We believe that this will make the paper much more accessible to
> a larger audience!
>
> > I believe that you could save some space e.g. by moving proofs in the
> > appendix, such that to include some additional figures to demonstrate
> > the steps of the approach.
>
> Thank you, we are revising the manuscript accordingly. While we believe
> that the proofs provide additional depth about the method, we will use
> the space to provide a more intuitive description of our method as well
> as showcase more results on an additional data set. This experiment is
> currently running, and we will provide more information as soon as
> possible.
>
> ## Additional experiment
>
> > In high dimensions a lot of data are necessary in order to approximate
> > the data manifold. I appreciate the synthetic and the
> > MNIST/FashionMNIST experiments, but I think that some additional real
> > world data in high dimensions should be conducted.
>
> Thanks for this suggestion! We are running an additional experiment on
> a high-dimensional biological data set at the moment and will revise the
> manuscript accordingly.
>
> **Please let us know if there are any further questions in the meantime!**

---

### Official Review · Reviewer_ntAA · 2022-10-24

**Confidence:** 4
**Correctness:** 4
**Technical Novelty And Significance:** 3
**Empirical Novelty And Significance:** 4
**Recommendation:** 6

**Clarity, Quality, Novelty And Reproducibility:**

The proposal is a new point of view and has a high degree of novelty and originality. The content also seems accurate, but more practical verification and suggestions for usage are needed to make it appropriate for the scope.

**Strength And Weaknesses:**

The issues and methods proposed are fine and appear to be theoretically sound. On the other hand, the concern is whether it is appropriate in terms of AI papers. Validation with real-world data has also been done, but it observes the relationship between each data class and indicator, which is different from the original goal of detecting singularities. Therefore, we do not know if it can be used for general AI tasks such as building AI models.For example, monitoring changes in the proposed index against data interpolation (transitions between two images) performed in the generative model, and comparing it to the classifier's confidency. Other considerations might include noise and adversarial attack detection.

Minor point, but the formatting seems incorrect. I don't know why the line numbers are attached, but they should be removed.

**Summary Of The Paper:**

This paper provides a method for detecting singularities in the data distribution in terms of the manifold hypothesis. The validity of the method is mathematically proven. Finally, it is applied to artificial data and real data to verify its effectiveness.

**Summary Of The Review:**

The problem setting is interesting and important, and the report presents a theoretical solution to the issue. On the other hand, its practical use is unclear and I question whether it is appropriate for the scope of the conference. While I believe it is still within the scope at this point, I think it should be in a form that does not raise concerns.

**Conclusion following discussion with the authors**

I believe that our biggest concern, contribution to the AI field, has been resolved by clearly showing that the manifold hypothesis detects situations in which it does not work. Since the specific application of the method remains to be determined, and since there was no score of 7 in the peer review process, the score will be stayed.

---

> ### Author Response · Authors · 2022-11-08
> **Our initial response to your comments**
>
> Thank you very much for your review and the related suggestions. We are
> glad to have captured your interest in our project, and we are grateful
> for appreciating the **novelty** of the given approach, as well as the
> overall **quality** of the method. We believe that the suggestions for
> improvement provided by your review can easily be accommodated within
> the revision cycle, and we are currently preparing a new version of the
> manuscript.
>
> ## Clarifications
>
> > Validation with real-world data has also been done, but it observes
> > the relationship between each data class and indicator, which is
> > different from the original goal of detecting singularities.
>
> Thank you for appreciating the **validation on real-world data**! We
> would like to clarify that all experiments (including the ones that were
> conducted on real data sets) do not make use of any label or class
> information, but rather were done on the whole dataset (i.e.
> independently of the class labels). In this sense, the Euclidicity
> procedure works _unsupervised_ and reveals information about the local
> neighbourhood structure (w.r.t the whole dataset) of individual points.
> Therefore, the goal is still to detect singular points in the dataset
> (i.e. points that are arguably far from admitting a euclidean
> neighbourhood).
>
> Our **surprising result** is that low-Euclidicity images exhibit less
> complex (and more linear) structures (i.e. mostly digits of ‘1’s in the
> case of MNIST, and pants for FMNIST), and that high-Euclidicity images
> admit higher geometric complexity (in terms of structures in the images
> that are non-linear). This points towards the use of our method as
> a general way of assessing data complexity/anomalies. We aim to follow
> up on this line of reasoning in a future publication.
>
> ## Applications & Future work
>
> > Therefore, we do not know if it can be used for general AI tasks such
> > as building AI models.For example, monitoring changes in the proposed
> > index against data interpolation (transitions between two images)
> > performed in the generative model, and comparing it to the
> > classifier's confidency. Other considerations might include noise and
> > adversarial attack detection.
>
> We are very grateful for your suggestions regarding applications and we
> are happy to include these ideas into the discussion section. We are
> also currently running additional experiments on a biological data set
> (will revise the manuscript ASAP).
>
> We also agree that our method lends itself to be integrated into ML
> models, such as autoencoders, to make them aware of singularities. We
> plan on following such research directions in future work, and see the
> main purpose of this study to create a mathematically strong framework
> for detecting singularities in the first place. Our findings on MNIST
> and FMNIST suggest that Euclidicity is able to detect the complexity of
> data in an unsupervised fashion. Our method thus opens up the door for
> new ML workflows; for instance, we could use Euclidicity as a weight
> function that permits models to focus on samples that are less likely to
> be classified correctly. We plan on building on this work in subsequent
> publications and are very grateful for your suggestions!
>
> > On the other hand, its practical use is unclear and I question whether
> > it is appropriate for the scope of the conference.
>
> Thank you for raising this concern! We consider our contribution to be
> the first step towards an unsupervised singularity-aware representation
> learning procedure that is capable of revealing geometric information
> about the underlying data space; we thus believe that the paper is very
> well within the scope, seeing it as a way to lead to new improved
> unsupervised learning methods. We are currently revising our manuscript
> with your suggestions in mind and hope that this makes the potential of
> our method more clear to a larger audience!
>
> ## Minor points
>
> > Minor point, but the formatting seems incorrect. I don't know why the
> > line numbers are attached, but they should be removed.
>
> We attached the line numbers only for the convenience of the reviewers,
> e.g. to easily address questions that arise in a certain line. We want
> to emphasise that this does not increase the space for the content—the
> format and margins are left unchanged by this.
>
> **Please let us know if there are any further questions in the meantime!**

---

> > ### Comment · Reviewer_ntAA · 2022-11-11
> > **New questions following revised draft**
> >
> > Hi Authors,
> >
> > Thank you for your response and for correcting manuscript.
> >
> > Let me confirm a few things and clarify that the modifications have made things more confusing.
> >
> > I think the motivation for this paper is to figure out the parts that do not contain the manifold hypothesis, since manifold learning, etc. may not work well when the manifold hypothesis contains parts that do not hold. In this paper, the regions that do not satisfy the manifold hypothesis are called "singularity," and such regions have a reduced dimension as a manifold compared to the entire data, or the geodetic cannot be obtained appropriately. PID evaluates the former and Euclidicity evaluates the latter.
> >
> > On the other hand, is what is shown in the experiment discussed in terms of the above? On the other hand, are the results shown in the experiment discussed in terms of the above?
> > Including the results shown in the additional experiments, one would imagine that it would be possible to detect regions that do not satisfy the manifold hypothesis. However, it does not seem clear.
> > Table.1 is an analysis of data of almost constant dimension, and it is unclear whether it is possible to extract the singular area, and fig.5 and fig.6 discuss differently from the extraction of the singular area.(I admit that these are indirect evidence.) Can you discuss the purpose and results of these experiments in terms of the manifold hypothesis?
> >
> > Also, although we refer to the region we want to detect as singularity, this is relative to the entire data, so it is misleading to use a term that would lead people to perceive the detected data as problematic data. It is supposed to mean that the detected data is not convenient for methods such as manifold learning.
> >
> > In Figs. 4, 5, and 6, it seems that Euclidicity is higher where the density is higher in Figs. 4 and 5 ('1' in MNIST seems to have a higher density as a distribution), but Fig. 6 shows that Euclidicity is higher where the density is smaller. These are very confusing, but can you explain them in a simple way? (It is also confusing that Euclidicity and color correspond to opposite values in Figs. 4 and 6.)
> > Also, if the discussion is only about the size of the density, I am concerned about the difference from Gaussian density, etc.
> >
> > Best,

---

> > > ### Author Response · Authors · 2022-11-11
> > > **Thanks & some clarifications**
> > >
> > > First of all: **thank you very much for responding to our revision—we really appreciate the time and the comments!**
> > >
> > > Please accept our sincere apologies for the additional confusion! We tried to be "terse" & discuss new aspects within the page limits, but following this discussion round, we will rewrite the experimental section in accord with all reviewers. To this end, we have some questions to you, which we mention in a separate response.
> > >
> > > > I think the motivation for this paper is to figure out the parts that do not contain the manifold hypothesis, since manifold learning, etc. may not work well when the manifold hypothesis contains parts that do not hold. In this paper, the regions that do not satisfy the manifold hypothesis are called "singularity," and such regions have a reduced dimension as a manifold compared to the entire data, or the geodetic cannot be obtained appropriately. PID evaluates the former and Euclidicity evaluates the latter.
> > >
> > > To provide one more detail: our singularity score has to be seen _relative_ to other points in the data set.
> > >
> > > > [...] On the other hand, are the results shown in the experiment discussed in terms of the above? Including the results shown in the additional experiments, one would imagine that it would be possible to detect regions that do not satisfy the manifold hypothesis. However, it does not seem clear. Table.1 is an analysis of data of almost constant dimension, and it is unclear whether it is possible to extract the singular area and fig.5 and fig.6 discuss differently from the extraction of the singular area. [...] Can you discuss the purpose and results of these experiments in terms of the manifold hypothesis?
> > >
> > > Thank you for raising this! We argue that we _are_ detecting such regions, but the experiments tackle different nuances of this: Table 1 refers to an experiment with _different_ dimensions; it needs to be contextualised with Fig. 10: here, we are gluing a circle to a sphere. The gluing point is a singularity by our definition. Table 1 shows that we correctly detect the fact that the dimensionality of neighbourhoods _changes_ as one approaches the gluing point. When looking at points on the circle (1D), we see that the _mean_ dimension is greater than 1, indicating that not all points admit a 1D neighbourhood. Likewise, for the sphere, we see that not all points afford a 2D neighbourhood, since the mean dimensionality is (slightly) lower than 2 if one approaches the gluing point. Our PID score detects this fact correctly (as shown in Figure 10). This space is thus **not** a manifold on its own, but consists of manifold parts.
> > >
> > > Figure 5 shows examples of a similar phenomenon: data points that are well described by the intrinsic dimension _relative_ to the rest of the data set appear to have lower "geometric" complexity than others since these points are well described by "linear", i.e. very simple, neighbourhoods (a neighbourhood that contains only the digit `1` is easier to characterise than a nb'hood containing other digits).
> > >
> > > > Also, although we refer to the region we want to detect as singularity, this is relative to the entire data, so it is misleading to use a term that would lead people to perceive the detected data as problematic data. [...]
> > >
> > > This is why we use the term "singularity", because such regions constitute _potentially problematic_ regions _if_ one assumes—like many manifold learning algorithms do—that the structure of the space is homogeneous. Our definition coincides with the mathematical definition of singularity, since we are able to handle stratified spaces, for instance (in fact, we believe that stratified spaces may provide a much better description of complex data sets in practice; this prompted the development of TOAST in the first place).
> > >
> > > > In Figs. 4, 5, and 6, it seems that Euclidicity is higher where the density is higher in Figs. 4 and 5[...], but Fig. 6 shows that Euclidicity is higher where the density is smaller. These are very confusing [...] (It is also confusing that Euclidicity and color correspond to opposite values [...]) Also, if the discussion is only about the size of the density, I am concerned about the difference from Gaussian density, etc.
> > >
> > > We have fixed the colour maps now—sorry about the confusion (red/dark now always indicates high Euclidicity scores). The result shows actually a strength of our method: while the "pinch point" and "gluing point" are singularities that _can_ be detected via local density, Euclidicity can also detect other types of singularities. This is what is shown in Fig. 6. In general, Euclidicity provides complementary information to density: density is highlighting the spatial/geometric distribution of points, while Euclidicity makes statements about their relationship to the rest of the space. (We are not sure about the comment concerning density in MNIST; we never measured this information; Fig. 15 shows Euclidicity distributions).
> > >
> > > We hope this clarifies your concerns!

---

> > > ### Author Response · Authors · 2022-11-11
> > > **Additional questions for the reviewer**
> > >
> > > Again, we would like to thank  the reviewer for engaging so positively with our work! We have some minor questions about how to best restructure our revision.
> > >
> > > Since we want to provide additional clarity about the experiments, this is the revised experimental section we had in mind:
> > >
> > > 1. Parameter selection
> > > 2. Euclidicity captures singularities of synthetic data sets (plus a discussion of stability properties)
> > > 3. The multi-scale approach of Euclidicity is crucial for obtaining good results.
> > > 4. Euclidicity captures geometric complexity of high-dimensional image data sets
> > > 5. Euclidicity captures lower-dimensional structures in cytometry data
> > >
> > > We find that this structure would make the idea of how different regions are being captured much more precise (notice the absence of the PID experiment, which we would put in the appendix again), while at the same time providing a demonstration of the stability properties of our method.
> > >
> > > *Would this be a suitable structure to alleviate your concerns?*
> > >
> > > We could also optionally put experiment 4 in the appendix (even though we are convinced of its utility for the main paper) and thus delineate more space to the other experiments.
> > >
> > > Following our discussion, we seem to agree on the "technical" description of the method, and our revision appears to be mostly concerned with matters of framing our content in light of some demonstrations of various representation learning scenarios in which Euclidicity might be useful.
> > >
> > > Thanks for engaging with our submission!

---

> > > > ### Comment · Reviewer_ntAA · 2022-11-14
> > > > **Additional comment**
> > > >
> > > > Thank you for considering the correction.
> > > >
> > > > It is not appropriate for me as a reviewer to comment on the details of the revisions, so I will describe the concerns that I would like to see resolved.
> > > >
> > > > Basically, the experiments should show that the proposed method is effective for the issue focused on. Of course, there is no problem if it shows additional effects. The concern is that it is difficult to know what the paper is about because it is written in a nuanced way that is different from the issue the experiment is focused on. I think this is why reviewers, including myself and others, find it difficult to see the effectiveness of this paper as an AI. At the very least, the relationship between each experiment and the objective of the paper, i.e., the detection of singularity for the manifold hypothesis, should be clearly stated. I think the lack of explanations to be clearly stated, including about the word "singularity," tends to lead to misunderstandings.
> > > >
> > > > Best,

---

> > > > > ### Author Response · Authors · 2022-11-14
> > > > > **New summaries**
> > > > >
> > > > > Dear reviewer,
> > > > >
> > > > > Thank you once more for your time and for engaging with our work!
> > > > >
> > > > > Prior to summarising the experiments, we want to provide a brief explanation of our definition of *singularity*:
> > > > > **In our framework, singularities are points that do *not* have a Euclidean neighbourhood in the intrinsic dimension of the space.**
> > > > >
> > > > > For example, the 'pinch point' constitutes a singularity because it does not admit a locally two-dimensional neighbourhood (all other points of the pinched torus are exactly like points on a torus, which can be locally described by two coordinates).
> > > > >
> > > > > We now continue with our comments.
> > > > >
> > > > > > Basically, the experiments should show that the proposed method is effective for the
> > > > > > issue focused on.
> > > > >
> > > > > We fully agree! Our experiments were designed to show that the method does what it is intended to do; we will try and clarify this and make the connection between the experiments and this issue statement more clear. The issue we focus on is two-fold: firstly, we want to devise a method in order to detect the intrinsic dimension of data. We do this with a multiscale method called PID. Secondly, we want to devise a way to find points which are not on the same underlying manifold as the rest of the data. We therefore introduce Euclidicity, which serves as a tool to measure the deviation for a point to have a Euclidean neighbourhood. (We leave the more technical details to Section 4).
> > > > >
> > > > > > The concern is that it is difficult to know what the paper is about because it is written in a > nuanced way that is different from the issue the experiment is focused on.
> > > > >
> > > > > Let us try to clarify why we believe that the experiments are in line with the issues we focus on.
> > > > >
> > > > > - Experiment 5.2 shows that PID is capable of detecting the correct dimension when the data is comprised of several different dimensions (here, a 1D circle and a 2D square). We find our method is best able to detect the mixed-dimensionality, which confirms our hypothesis that PID can detect the true intrinsic dimension of the data.  This is important since PID is based on new concepts (local (persistent) homology) as opposed to other methods.
> > > > >
> > > > > - Experiment 5.3 shows that Euclidicity is able to detect singularities in the sense of our definition of stratified simplicial complexes. We use synthetic data sets because we want to make sure that we ‘catch’ the right type of singularity, and we find that Euclidicity is able to correctly identify these singular points (the dark red points in Figure 4)
> > > > >
> > > > > Experiments 5.2 and 5.3 were the main experiments to test whether PID and Euclidicity address the issues we introduce. Experiments 5.4--5.6 are now intended to deepen our understanding of our proposed methods:
> > > > >
> > > > > - Experiment 5.4 discusses the multiscale aspect of our approach, where we find that our method outperforms one-parameter approaches. This is relevant because a fair objection to our work could be that multiple parameters are not required in practice.
> > > > >
> > > > > -  After the encouraging results showing the promise of our method, we ran Experiment 5.5 to assess the Euclidicity of benchmark ML datasets. We observe that high-Euclidicity regions (i.e. singular regions) have higher geometric complexity in the underlying images, which leads us to hypothesise that our method has the potential to become an novel expressive unsupervised representation learning procedure.
> > > > >
> > > > > - Experiment 5.6, which we added in the rebuttal to address your feedback on real-world applications, shows the Euclidicity procedure on a real-world biological dataset. Here, we observe that lower-dimensional subsets (relative to the overall dimension of the input data) have higher-Euclidicity scores (i.e. should be considered as 'more singular'). Again, this is in alignment with our definition of singularities via stratified simplicial complexes: in this case, the stratification has to be taken along the distinct dimensions.
> > > > >
> > > > > To summarise, 5.2 shows effectiveness of PID (the first novel concept we focus on), and 5.3 and 5.4 show effectiveness of Euclidicity (the second novel concept we focus on). Finally, in 5.5 and 5.6 we interpret singular regions in real-world data by making use of our approach.
> > > > >
> > > > > If you agree with these clarifications and find them helpful, we are happy to integrate them in a new revision! Please let us know if you have any further questions!

---

### Official Review · Reviewer_8kd2 · 2022-10-26

**Confidence:** 2
**Clarity, Quality, Novelty And Reproducibility:** Poor clarity, OK in quality and novel…
**Correctness:** 4
**Technical Novelty And Significance:** 3
**Empirical Novelty And Significance:** 2
**Recommendation:** 3

**Strength And Weaknesses:**

Strengths

- To the best of my non-expert understanding the formulation appears to be novel. I also did find the arguments highlighting the scenarios of “singularities” in the data to be interesting
- Most experiments do demonstrate the characteristics of the claimed contributions, i.e. the measures PID and Euclidicity provide an interesting way of measuring the complexity of data, with singularities showing up nicely (especially in Figure 1 and 5).
- The appendix demonstrates some interesting experiments, especially Figures 8 and 10, where a comparison is shown between some well known dimensionality estimators and the proposed approach.

Weaknesses and Questions

- Despite the focused approach to handling singularities in data, I do feel the overall applicability of this method in practice is not well explored. Most experiments are very synthetic and tailor-made for the setting of this paper (eg. pinched torus, wedged spheres etc.)
- The structure of the paper can be made much better. I feel a lot of the important practical results are left in the appendix. For example sections, A5, A6, and A7 should (with appropriate editing) belong to the main paper and must be argued as a core part of the contribution.
- Unfortunately, I did not find the material to be satisfactorily self-contained, and I think that it would go a long way to make sections 2 and 4 more accessible to a general representation learning audience.
- (suggestion) Consider improving the quality of visualizations, in terms of choosing the view in 3D (eg Figure 10, to see the point of contact between the circle and sphere).
- What is the limit on line 111 ?


**Summary Of The Paper:**

This paper proposes two measures that encapsulate the topological features of data: Persistent Intrinsic Dimension (PID) and Euclidicity. The authors claim that manifold learning with a fixed intrinsic dimension can be a restrictive assumption in general and it is worth thinking of allowing this dimension to vary for different points on the manifold. The seeming advantages are argued to be the ability to identify singularities (i.e. points that violate the manifold assumption) as well as providing a notion of local geometric complexity in the data.

The authors spend considerable effort in theoretically elaborating their approach and providing theorems to show that their persistent homology-based method is stable to choice of parameters and sampling issues in the data. Synthetic demonstrations are made on the pinched torus, the wedges spheres as well as image datasets like MNIST and FMNIST.


**Summary Of The Review:**

Overall, I think this paper proposes an interesting contribution in topological data analysis. However, I am leaning negative considering two factors (1.) The overall confusing structure of the paper and inaccessibility of the core TDA components (2.) Very little emphasis on actual applications with significant impact.

---

> ### Author Response · Authors · 2022-11-08
> **Our initial response to your comments**
>
> Thank you very much for your review and the related suggestions. We are
> glad to have captivated your interest in our project, and we are
> grateful for appreciating the **novelty** of the given approach, as well
> as its usefulness for measuring the **complexity** of data. We believe
> that the suggestions for improvement provided by your review can easily
> be accommodated within the revision cycle, and we are currently
> preparing a new version of the manuscript.
>
> Meanwhile, please find our answers to your comments below:
>
> ## Dimensionality estimation comparison
>
> > The appendix demonstrates some interesting experiments, especially
> > Figures 8 and 10, where a comparison is shown between some well known
> > dimensionality estimators and the proposed approach
>
> Thanks! We agree that the PID experiment in the appendix highlights the
> utility of the proposed dimension estimation procedure. We would like to
> emphasise that the **core** of our work is dedicated to the detection of
> singular regions in data via Euclidicity, as well as the geometric
> interpretation of such regions for high-dimensional data. We will
> summarise the PID experiment in the main text now.
>
> ## Overall applicability
>
> > Despite the focused approach to handling singularities in data, I do
> > feel the overall applicability of this method in practice is not well
> > explored. Most experiments are very synthetic and tailor-made for the
> > setting of this paper (eg. pinched torus, wedged spheres etc.)
>
> For this paper, our main goal is to provide a fundamental study of the
> guarantees and limitations of the proposed method, which is necessary to
> justify and assess its practical utility in the first place. Being the
> **first investigation** of singularity structures in data, we had to
> create synthetic datasets with known singularities (serving as a ground
> truth) to be able to assess the practical performance of the method.
>
> We are running **additional experiments** at the moment (will provide
> updates ASAP). We also consider our method to open the door towards
> other approaches in the future, but we feel that the **primary aim** of
> this paper lies in providing a theoretically concise and mathematically
> rigorous framework. We will revise our paper to discuss potential
> application areas in more detail, in particular the integration of our
> method into ML models to make the ‘singularity-aware’; we consider such
> integrations to require substantial additional research though, which is
> best tackled in a subsequent publication.
>
> ## Clarity
>
> > The structure of the paper can be made much better. I feel a lot of
> > the important practical results are left in the appendix. For example
> > sections, A5, A6, and A7 should (with appropriate editing) belong to
> > the main paper and must be argued as a core part of the contribution.
>
> Thanks for this suggestion! We will restructure the paper accordingly
> and update you once we have a revision. The main intent of the appendix
> is to show reproducibility of the results, and to quantitatively
> underscore the respective qualitative statements about Euclidicity
> scores in the main text. We will revise the paper to include additional
> information about the persistent intrinsic dimension (PID) in the main
> text.
>
> > Unfortunately, I did not find the material to be satisfactorily
> > self-contained, and I think that it would go a long way to make
> > sections 2 and 4 more accessible to a general representation learning
> > audience.
>
> Thanks for this suggestion! We will provide more high-level explanations
> to make the method more accessible; we also appreciate your comments re:
> our visualisations and will improve them accordingly. We will upload
> a revised version ASAP.
>
> > What is the limit on line 111 ?
>
> Intuitively, this is supposed to represent homology (i.e. topological
> features) in an infinitesimally small punctured neighbourhood around
> a point (this is tantamount to looking at the space without considering
> the given point). Formally, the limit on line 111 is a categorical
> colimit (we provide more notational details in A.1, but these are not
> required for a high-level understanding of our method). Heuristically,
> an element in this colimit object is given by a homology class in
> $H_i(X,X \setminus U)$, for $U$ being an infinitesimal small
> neighbourhood of $x$.
>
> **Please let us know if there are any further questions in the meantime!**

---

### Author Response · Authors · 2022-11-09
**First revision in light of reviewer comments**

Dear reviewers,

Once again thank you very much for your helpful comments! We are happy to present you with the first revision of our paper, in which we provided the following changes (large-scale updates are marked in blue):

- We have expanded on **clarity and intuition** behind the methods
- We have updated Figure 2 to make it easier to understand
- We provided a more **intuitive overview** of the background of our method
- We have provided a brief introduction to our methods in Section 4
- We have restructured Section 4 and compressed several technical details; in particular, we moved proofs to the appendix now.
- As a **new experiment**, we describe the comparison with the intrinsic dimension estimator in Section 5.2
- For clarity, we merged two of the wrapped figures and created Figure 4, which discusses "wedged spheres"
- We have extended the discussion to comment on **future research directions**, in particular as it concerns singularity-aware ML models
- We improve Figure 9 (in the new numbering; prev. Figure 10) in the appendix to use a different perspective

Please note that another large-scale experiment on a real-world data set (cytometry data) is still running. We will incorporate those results ASAP and provide another revision of the paper—stay tuned!

In the meantime, we want to thank you for your feedback on the manuscript.  We are happy to answer any further questions; should you be satisfied with our responses to your comments and concerns so far, we would be grateful if you considered reflecting this in your score.

---

### Author Response · Authors · 2022-11-11
**Second revision in light of reviewer comments**

Dear reviewers,

We are excited to present the second revision in light of your comments.
The major extension over the first revision is a **new experiment** on
a real-world cytometry data set, in which Euclidicity highlights the
existence of lower-dimensional substructures along so-called *branching
points*, i.e. points in which cells diverge into different lineages.
Without Euclidicity, special dimensionality reduction algorithms, which
are hard to train and verify, need to be used to characterise such
spaces. We find that this experiment provides ample evidence for
employing Euclidicity as a general tool in unsupervised data analysis,
guiding the exploration and making models singularity-aware.

# List of changes

- We added Section 5.6, a new section describing a **new cytometry experiment**.
- We added Supplementary Section A.9 with **additional details** about the new experiment, including Figure 17 (detailing intrinsic dimensionality estimates) and Figure 18 (demonstrating stability properties of Euclidicity for detecting relevant low-dimensional regions in the data set).
- We ensured that our content remains within 9 content pages. We are
  happy to restructure content according to further suggestions.

We believe that this second revision addresses your concerns, and
would like to thank you once again for the feedback on our manuscript.
We are happy to answer any further questions that may arise while
checking the revisions.

**In case you are satisfied with our responses to your comments, we would
be grateful if you considered reflecting this in your score.**

---

### Author Response · Authors · 2022-11-11
**Third revision in light of reviewer comments**

For the third revision, we only changed the colour map for Fig. 6 and Fig. 18 to be consistent with the remaining colour maps, where red/dark always indicates high Euclidicity. We thank reviewer `ntAA` for raising this issue.

---

### Author Response · Authors · 2022-11-16
**Further questions?**

Dear reviewers,

Thanks for engaging with our work! We are particularly grateful for `ntAA` for already engaging with us and commenting. In light of the approaching deadline for rebuttals, please let us know if there are any other questions. Moreover, if you find our updates to be satisfactory, we would kindly ask you to reflect this in our scores.

---

### Decision · Program_Chairs · 2023-01-20

**Decision:**

Reject

**Justification For Why Not Higher Score:**

While two reviewers recommend weak accept, one of them has low confidence while the other questions the relevance for AI.

**Justification For Why Not Lower Score:**

N/A

**Metareview: Summary, Strengths And Weaknesses:**

Summary:
This paper suggests two measures to quantify the topological properties of the data manifold: Persistent intrinsic dimension and Euclidicity.

Strengths:
- The formulations are novel and appear to nicely quantify singularities.

Weaknesses:
- The applicability of the metrics remain unclear, as the experiments are mostly synthetic and proof of concept in nature.
- In particular, the relevance for AI tasks was not clear to the reviewers.
- The paper was found a bit hard to read, and not be self contained.

In short, the reviewers appreciate the novel approach to detecting singularities in data. However, if the authors seek to publish at a mainstream machine learning conference, I recommend that even more care is taken to make the paper widely accessible and highlight the potential impact for mainstream AI to benefit from their work.